# Comparison of Mandibular Arch Expansion by the Schwartz Appliance Using Two Activation Protocols: A Preliminary Retrospective Clinical Study

**DOI:** 10.3390/jfmk5030061

**Published:** 2020-08-06

**Authors:** Vincenzo Quinzi, Stefano Mummolo, Francesca Bertolazzi, Vincenzo Campanella, Giuseppe Marzo, Enrico Marchetti

**Affiliations:** 1Department of Life, Health & Environmental Sciences, Postgraduate School of Orthodontics, University of L’Aquila, 67100 L’Aquila, Italy; stefano.mummolo@univaq.it (S.M.); fra.bertolazzi@gmail.com (F.B.); giuseppe.marzo@univaq.it (G.M.); enrico.marchetti@univaq.it (E.M.); 2Department of Clinical Science and Translational Medicine, University of Rome “Tor Vergata”, 00133 Rome, Italy; vincenzo.campanella@uniroma2.it

**Keywords:** tooth size-arch length discrepancy, mandibular arch expansion, mandibular crowding reduction, schwartz appliance, early interceptive orthodontic treatment

## Abstract

Background and objectives: Dental crowding is more pronounced in the mandible than in the maxilla. When exceeding a significant amount, the creation of new space is required. The mandibular expansion devices prove to be useful even if the increase in the lower arch perimeter seems to be just ascribed to the vestibular inclination of teeth. The aim of the study was to compare two activation protocols of the Schwartz appliance in terms of effectiveness, particularly with regard to how quickly crowding is solved and how smaller is the increasing of vestibular inclination of the mandibular molars. Materials and Methods: We compared two groups of patients treated with different activation’s protocols of the lower Schwartz appliance (Group 1 protocol consisted in turning the expansion screw half a turn twice every two weeks and replacing the device every four months; Group 2 was treated by using the classic activation protocol—1/4 turn every week, never replacing the device). The measurements of parameters such as intercanine distance (IC), interpremolar distance (IPM), intermolar distance (IM), arch perimeter(AP), curve of Wilson (COW), and crowding (CR) were made on dental casts at the beginning and at the end of the treatment. Results: A significant difference between protocol groups was observed in the variation of COWL between time 0 and time 1 with protocol 1 with protocol 1 subjects showing a smaller increase in the parameter than protocol 2 subjects. The same trend was observed also for COWR, but the difference between protocol groups was slightly smaller and the interaction protocol-by-time did not reach the statistical significance. Finally, treatment duration in protocol 1 was significantly lower than in protocol 2. Conclusion: The results of our study suggest that the new activation protocol would seem more effective as it allows to achieve the objective of the therapy more quickly, and likely leading to greater bodily expansion.

## 1. Introduction

Dental crowding is the most common form of malocclusion. Over the years many studies have investigated how it affects a large part of the population [1,2] and found that, if left untreated, it unavoidably worsens [2,3,4]. Although the role of environment and genetics in the etiology of dental crowding is still questioned, new evidence has emerged that indicates that dental crowding is a malocclusion common to modern postindustrial human populations, occurring as a result of increased processing of modern foods. [5,6]

Crowding depends on the relationship between the size of the teeth and the dimension of dental arches [3,7,8]. It normally affects both the maxilla and the mandible even if, during life, it is destined to worsen more in the lower arch regardless of having undergone or no an orthodontic treatment performed within well accepted guidelines [3,5,9].

From an orthodontic point of view, since during the transition from mixed to permanent dentition we lose arch length, it is clear that an early treatment of dental malalignment is highly recommended. As long as crowding does not exceed the size of leeway space, it will be possible, especially in the lower arch, to manage it by maintaining the arch length during the transition period [10]. Greater crowding will require the creation of new space. It is well known nowadays, that the rapid maxillary expander (RME) is the most effective device to expand a constricted maxillary arch and to increase the dental arch perimeter [11,12,13,14]: depending on the activation protocol chosen according to patient needs, it allows to obtain more dental or skeletal effects (both in transverse and sagittal dimension) [14,15]. On the other hand the expansion of the lower arch has always been a controversial issue: since the mandible does not have the midpalatal suture, true orthopedic expansion is unlikely, unless recently developed distraction osteogenesis techniques are used. It has been accepted that the effects gained with devices such as the Schwartz appliance (Figure 1) or the lip-bumper could be just ascribed to an increase in dental inclination or, at most, to a phenomenon related with the alveolar bones [11,12,13,14].

The aim of this study was to compare two lower Schwartz appliance activation protocols to understand which one is more effective in terms of time needed to reduce mandibular crowding, of reduction of mandibular crowding and which one allows a more bodily expansion.

The null hypothesis was that the two protocols were equally effective and that both determined an increase in the lower arch perimeter mainly due to the vestibular inclination of teeth.

## 2. Materials and Methods

### 2.1. Patients

The present protocol was approved by the Ethics Committee of the University of L’Aquila (Document DR206/2013, 16 July 2013). This retrospective clinical study was performed by comparing the dental casts of all the patients that underwent orthodontic treatment in two private clinics in the period between November 2016–October 2019. This study was approved by the Clinical Research Ethics Committee (CEIC), following the principles of Helsinki for human experimentation. The group 1 was composed of 12 patients (5 males and 7 females, 41.7% and 58.3%, respectively) whose average age was 8.1 ± 0.5. The group 2 was initially composed of 13 patients (7 males and 6 females) of 7.8 ± 0.7 years of age but two patients were excluded due to the lack of collaboration. Therefore, group 2 was composed of 11 patients (5 males and 6 females, 45.5% and 54.5%, respectively). Being an observational study, patients had not been randomly assigned to one group or to the other. We have selected all the patients who consecutively followed the two different protocols in the two clinics and who received the treatment in the pre-established period time. Cephalometric measurements were not performed since they would not have provided useful information for the study purpose.

The inclusion criteria were the following:-Class 1 malocclusion-Mixed dentition-Moderate to severe crowding (−4 mm or more)-Age at the beginning of the treatment < 9 years-Exclusion criteria were:-Posterior cross-bite-Previous orthodontic treatment-Dental agenesis-Systemic syndromes

### 2.2. Measurements of Dental Cast

The parameters were measured on dental casts at the beginning of the treatment (T0) and at the end (T1) by the same operator with a manual caliber to eliminate interexaminer errors. The operator was blind with respect to the treatment received by patients in order to prevent bias in measurements. Moreover, measurements were repeated one month later by the same operator in order to avoid intraexaminer errors. The intraclass correlation coefficient (ICC) computed on the paired measurements was always greater than 0.98 (from 0.986 to 0.999).

The measurements performed concerned the following parameters: intercanine distance (IC), interpremolar distance (IPM), intermolar distance (IM), arch perimeter (AP), curve of Wilson (COW), crowding (CR) (Figure 2 and Figure 3).

The IC and IPM distance were assessed as the distance between the inner lingual points of the gingival margin of permanent or deciduous canines and of the first premolars or first deciduous molars. The IM was defined as the distance between the point of intersection of the lingual groove with the cervical gingival margin of the first molars [16].

The arch perimeter was instead measured as the sum of four segments:-segment (a): from the mesial aspect of the right first permanent molar to the distal aspect of the right permanent or deciduous canine-segment (b): from the distal aspect of the right permanent or deciduous canine to the contact point between the permanent central incisors-segment (c): from the contact point between the permanent central incisors to the distal aspect of the left permanent or deciduous canine-segment (d): from the distal aspect of the left permanent or deciduous canine to the mesial aspect of the left first permanent molar.

CR was calculated as the discrepancy between the mesio-distal dimension of teeth and the length of the segment taken into consideration to calculate the arch perimeter.

COW was measured with a protractor as the angle between the occlusal plane (defined by a ruler resting on the vestibular cusps of the inferior first permanent molars) and a tangent to the vestibular aspect of the first permanent molars. We measured both the right COW (COWR) and the left COW (COWL).

### 2.3. Treatment Protocols

Patients belonging to both groups underwent a treatment of maxillary and mandibular expansion with upper and lower Schwartz appliances. The devices were built with acrylic resin, ball hooks and Adams hooks in stainless steel (which thickness was 0.8 mm and 0.7 mm respectively) and an expansion screw. The appliances were “activated” by turning this screw. The protocol followed by group 1 involved the doctor activating the appliance of 2/4 turn every 15 days and the replacement of the device every 4 month during all the treatment period even if the screw had not reached its expansion limit. The replacement was made to allow the patient to always wear a device with a perfect fitting, built every 4 months starting from a new impression that faithfully reproduced the progressively changed situation. The group 2 protocol required the activation of the appliance of a 1/4 turn per week by the parents of the patient and the use of the same device for the whole period unless the screw had reached its expansion limit before the end of the treatment. This occurred once in 2 patients out of 11. Patients were instructed to wear the device for a minimum of 14 h per day (Figure 4).

### 2.4. Statistical Methods

Categorical variables are presented as absolute and percent frequencies. Quantitative data are summarized by means, standard deviations (SD), 95% Confidence Intervals (CI), medians and ranges; Cohen’s d is used as quantitative measure of the magnitude of the difference between the means of the two protocols [17]. Wahlsten (2011) suggested d = 0.5, 1.0 and 1.5 as reference values for small, medium and large effect sizes [18].

Quantitative variables were analyzed by parametric mixed-model analysis of variance, with treatment and sex as grouping factors and time as repeated measures. In the presence of significant interactions (treatment-by-sex, treatment-by-time, sex-by-time or treatment-by-sex-by-time), multiple comparisons were performed by Student t test with Bonferroni correction. Statistical analyses were performed by STATA 16.0.

The sample size of 11 vs. 12 subjects allowed us to assess non inferiority of Protocol 1 vs. Protocol 2, with pairwise alpha = 0.0125 (corresponding to an experimentwise alpha = 0.05 with Bonferroni’s correction for 4 comparisons), power = 0.80 and non-inferiority limit = 1.3 SD (corresponding to a medium-large effect size according to Wahlsten 2011) [18]. Sample size calculation was performed using Sealed Envelope Ltd. 2012. Power calculator for continuous outcome non-inferiority trial. [Online] Available from: https://www.sealedenvelope.com/power/continuous-noninferior/ (Accessed on Monday 27 Jan 2020).

## 3. Results

Results are showed in Table 1 (description of the variables in the two protocol groups) and in Figure 5 (box and whisker plots showing parameter variation between T1 and T2 in different protocols). A significant increase in mean values between time 0 and time 1 was observed for all the variables (main effect of time always *p* < 0.001). No significant differences between males and females were observed for any variable. IPM significantly differed between group 1 and group 2 subjects (main effect of protocol: F(1,19) = 4.38, *p* = 0.05), more markedly at baseline and slightly less at the end of treatment, but the difference between treatment groups in the variation of the variable from time 0 to time 1 was not significant (interaction protocol-by-time: F(1,19) = 0.15, *p* = 0.70). On the contrary, a significant difference between protocol groups was observed in the variation of COWL between time 0 and time 1 (interaction protocol-by-time: F(1,19) = 6.01, *p* = 0.02), with protocol 1 subjects showing a smaller increase in the parameter than protocol 2 subjects. Indeed, protocol 1 and protocol 2 subjects did not differ with respect to COWL at time 0 (44.0 ± 4.7 and 44.4 ± 6.8 for protocol 1 and protocol 2, respectively), while they differed significantly at time 1 (52.2 ± 7.0 and 59.1 ± 6.8 for protocol 1 and protocol 2, respectively; *p* < 0.05 with Bonferroni correction).The same trend was observed also for COWR, but the difference between protocol groups was slightly smaller and the interaction protocol-by-time did not reach the statistical significance (F(1,19) = 2.86, *p* = 0.11). Finally, treatment duration in protocol 1 was significantly lower than in protocol 2 (main effect of protocol: F(1,19) = 19.08, *p* < 0.001), while neither the main effect of sex nor the interaction protocol-by-sex were significant.

## 4. Discussion

The expansion of lower arch has always been a controversial issue: because of the anatomic structure of the mandible, it has been stated that the expansion that can be reached with devices such as the Schwartz appliance or the lip-bumper appliance could be just ascribed to an increase in dental inclination or, at most, to a phenomenon related with the alveolar bones [11,12,16].

In our study we selected two groups of patients with class I malocclusion treated with Schwartz appliances and measured on their dental casts parameters such as IC, IPM, IM, arch perimeter, crowding COWR and COWL with a manual caliber, since it has been stated by previous studies that there is no clinical significance of difference between measurements performed manually or digitally [19,20]. The appliances were activated with two different protocols: the group 1 activation protocol involved the dentist activating the appliance of 2/4 turn in his private clinic every 15 days and replacing it every 4 months. On the other hand, the group 2 appliances were activated by the parents of the patient of 1/4 turn every week at home and the device was not replaced during the whole treatment period unless it had reached its maximum expansion. Protocol 1 is therefore slightly more expensive because it involves replacing the device with a new one every 4 months but, at the same time, it requires less collaboration from the parents because they are not required to activate it every week. The aim of the study was to assess whether the two protocols were differently effective in terms of time needed to resolve crowding, of the resolution of crowding and whether one of them was more likely to produce a bodily expansion.

To our knowledge this is the first study to compare two different protocols of activation of the Schwartz appliance. To date, studies have been performed to evaluate the efficacy of a single protocol of this device in terms of lower arch expansion.

In 2010 and 2011 Tai et al. evaluated the efficacy of the Schwartz appliance on the increase of transverse dimension of both jaws in young and growing patients. In both studies patients were treated with an activation protocol corresponding to our protocol 2 (1/4 turn every week) with the difference that they were instructed to wear their device only at night (instead of a minimum of 14 h per day) and that measurements at T1 were made after about 9 months of retention following the end of the treatment.

In our study the analysis of the effects reached with the new protocol (protocol 1) led us to observe statistically significant differences in the variation of the COWL. With respect to the COWR the results follow the same trend even if they cannot reach the level of significance.

We have noticed that using the protocol 1 the vestibular inclination of the mandibular molars is smaller than the one obtained by applying the protocol 2. We cannot compare the efficacy of protocol 1 with other studies. However, we can compare the characteristics of our group treated with protocol 2 with those of patients already mentioned in other articles and treated with the same protocol. The starting conditions of the arch perimeter in patients observed in our study are similar to those of patients treated in the investigation by Tai et al. On the other hand, the starting conditions with respect to the amount of crowding are very different. Crowding is greater in patients involved in our study (the inclusion criteria stated that patients with crowding less than 4 mm had to be excluded): protocol 1 crowding_0–7.1 (SD 2.3), protocol 2 crowding_0–6.6 (SD 2.3) against crowding in the Tai et al. expanded group −3.59 (SD 1.21). This would explain why the increase in COWR and COWL is more in patients treated with protocol 2 (COWR 14.1 ± 6.6; COWL 14.7 ± 6.8) than in patients treated by Tai et al. (inclination of the right mandibular molars 8.5 ± 1.5; inclination of the left mandibular molars 8.9 ± 1.4): the greater the crowding, the more the correction of crowding, by using the Schwartz appliance activated with 1/4 turn every week, is up to proclination of teeth. However, we cannot underestimate that Tai et al. evaluated the molars inclination differently to us (using a CBCT): therefore, measurements are not perfectly superimposable.

Nowadays, another protocol used to correct this form of malocclusion involves the expansion of the maxilla with an RME and the mandibular expansion with a lower Schwartz appliance [13,14,21]. The only result obtained by Wendling et al., which is relevant to our investigation, is that the combined use of RME and lower Schwartz appliance prevent the mesial movements of the first lower permanent molar (−2 mm) if compared with the treatment performed with the RME alone. This can be useful if gaining arch length is needed [21].

Some authors suggest that in these cases it is more reliable to begin expanding the lower arch by using the Schwartz appliance to reach a dentoalveolar decompensation in order to have a more accurate reference during the maxillary expansion [13,14]. In the investigation of Grady et al. three groups of patients were compared: the first one was treated just with an RME, the second with an RME and a lower Schwartz appliance activated with a protocol corresponding to our protocol 2 and the third consisted of untreated patients. Since the authors do not report the starting conditions of crowding in these patients, we cannot compare this sample to ours. Moreover, our study groups were treated with both upper and lower Schwartz appliance. However, it is interesting to compare the modifications of the mandibular molar inclination which is 11° in the RME + Schwartz group of Grady et al. and 8.9 ± 6.7 (COWR) and 8.2 ± 4.9 (COWL) in our protocol 1 group.

We must consider that patients with the same amount of crowding may have a more or less pronounced COW: therefore, this may need a greater or smaller correction. The process of choosing the most suitable protocol can be suggested by the needs of each patient. At the same time, the selection of the most appropriate protocol can be dictated by the device used to expand the upper arch: for example, the protocol 1 can be more useful in those patients where the maxillary expansion is performed with the RME. This is because the Schwartz appliance activated with protocol 1 allows mandibular expansion to follow the rapid maxillary expansion more quickly and effectively (if it is not indicated to start expanding the lower arch and then the maxilla for any reason).

We are aware of the limitation of our study. Among these are the sample size of patients treated using both protocols. Moreover, the lack of a long-term follow up, of treatment does not provide us information about which protocol is more effective in the long-term perspective. This is a big limitation since many authors are skeptical about expanding the mandible permanently whereas other are not [22,23,24,25,26,27,28,29,30,31,32]. We should also measure the thickness of alveolar bone by using a CBCT to assess exactly where growth or bone resorption phenomena are located as Tai et al. did in their first investigation.

For these reasons, the results of our study must be interpreted with caution, although they suggest that both protocols work well with two substantial differences concerning time needed and the COW. However, if we interpret such results from a purely clinical point of view, it is clear that our protocol is more effective as it allows clinicians to obtain the desired results more quickly and with a small effort (just spending five minutes for a check-up every two weeks) by regularly monitoring the patient cooperation, the progress achieved and motivating them if needed.

Whisker represents the minimum and the maximum values of the distribution while the box represents the first, second (median) and third quartiles.

## 5. Conclusions

We introduced a new effective protocol that can be used to activate the Schwartz appliance. Further studies are needed on a larger sample of patients and with a long-term evaluation to reach more reliable conclusions on the efficacy of the treatment and on the duration of the effect. Nevertheless, this study provides information that can be applied to our daily clinical practice. It seems that the new protocol produces different effects with respect to the Curve of Wilson if compared with the one which have been used until now, so we can reject the null hypothesis of equal efficacy of the two protocols in terms of bodily expansion and time needed to reach a satisfying result. Moreover, it allows to reach our therapy goals in a shorter time. Every orthodontic case is different. This means that we can choose the most suitable protocol according to the clinical needs of every single patient.

## Figures and Tables

**Figure 1 jfmk-05-00061-f001:**
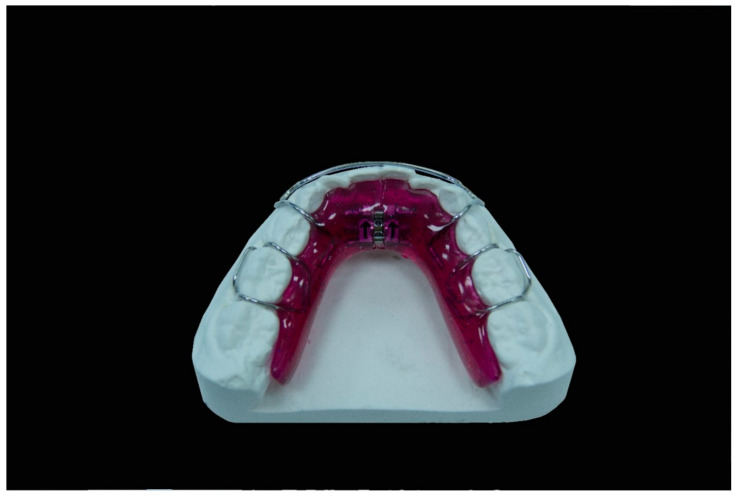
Schwartz appliance.

**Figure 2 jfmk-05-00061-f002:**
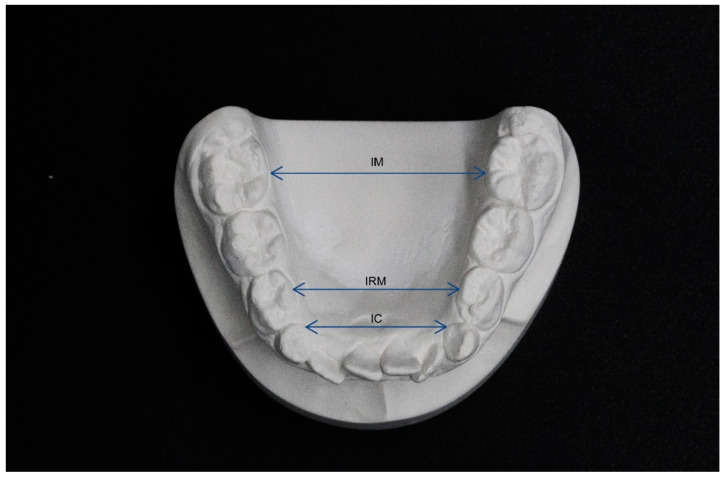
Measurements of dental cast: intermolar distance (IM—distance between the point of intersection of the lingual groove with the cervical gingival margin of the first molars), interpremolar distance and intercanine distance (IPM and IC—distance between the inner lingual points of the gingival margin of permanent or deciduous canines and of the first premolars or first deciduous molars.

**Figure 3 jfmk-05-00061-f003:**
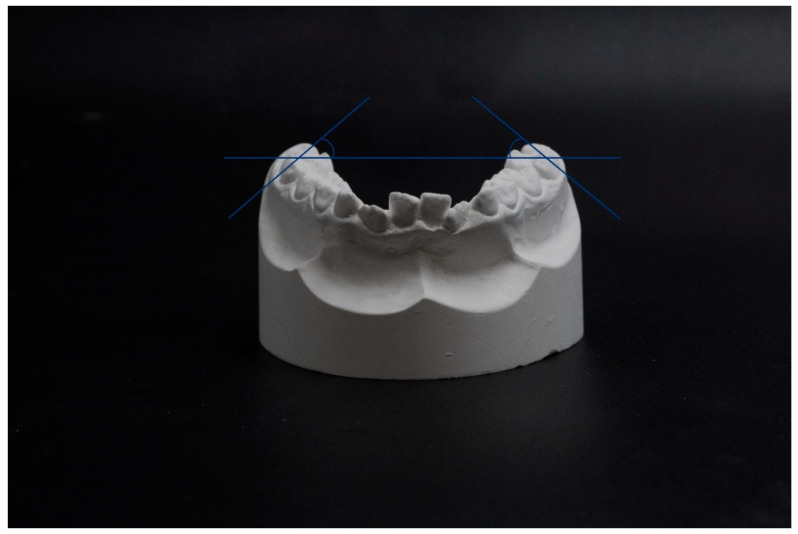
Curve of Wilson (COW—measured as the angle between the occlusal plane and a tangent to the vestibular aspect of the first permanent molars).

**Figure 4 jfmk-05-00061-f004:**
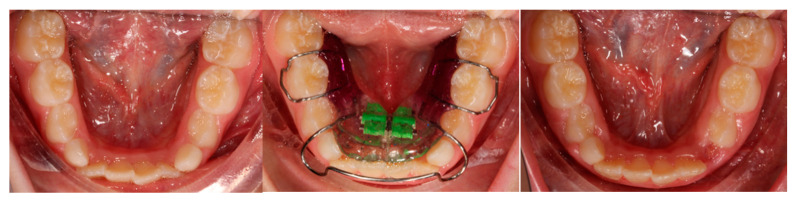
Clinical case: patient treated with the mandibular Schwartz appliance using protocol 1.

**Figure 5 jfmk-05-00061-f005:**
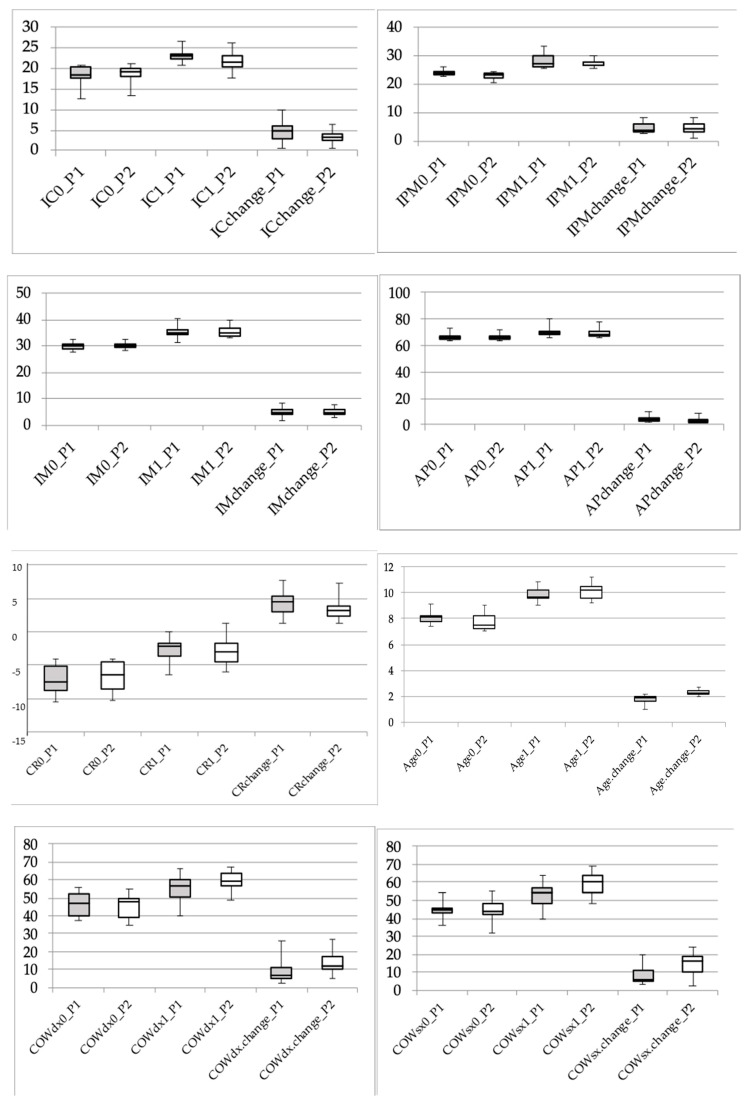
Box and whisker plots showing parameter variation between T1 and T2 in different protocols (P1 Protocol 1–P2 Protocol 2). IC: intercanine distance; IPM: interpremolar distance; IM: intermolar distance; COW: curve of Wilson, right R, left L.

**Table 1 jfmk-05-00061-t001:** Variables measured in children of the two protocol groups: descriptive values and significance levels in the analysis of variance (ANOVA).

	Protocol	1						Protocol	2						Effectsize		ANOVA’s *p*						
					95	%	CI					95	%	CI			Prot	Sex	P*S	Time	P*T	S*T	P*S*T
Variable	n	Mean	±	SD	Lower	;	Upper	n	Mean	±	SD	Lower	;	Upper	Cohen’s d								
**IC**																	0.639	0.764	0.170	**< 0.001**	0.139	**0.047**	0.524
**T0**	12	18.5	±	2.3	17.1	;	19.9	11	18.6	±	2.1	17.2	;	20.0	0.05	-							
**T1**	12	23.0	±	1.4	22.1	;	23.9	11	22.0	±	2.4	20.3	;	23.6	0.52	s							
***change***	*12*	***4.5***	***±***	***2.4***	***3.0***	***;***	***6.0***	*11*	***3.4***	***±***	***1.8***	***2.2***	***;***	***4.5***	*0.52*	*s*							
**IPM**																	**0.050**	0.463	**0.024**	**< 0.001**	0.699	0.856	0.301
**T0**	12	24.1	±	1.1	23.4	;	24.8	11	22.7	±	1.3	21.9	;	23.6	1.17	ml							
**T1**	12	28.5	±	2.5	26.9	;	30.1	11	27.4	±	1.4	26.4	;	28.3	0.54	s							
***change***	*12*	***4.4***	***±***	***2.0***	***3.1***	***;***	***5.7***	*11*	***4.6***	***±***	***2.1***	***3.2***	***;***	***6.0***	*0.10*	*s*							
**IM**																	0.842	0.772	0.087	**< 0.001**	0.861	0.930	0.192
**T0**	12	30.2	±	1.4	29.3	;	31.1	11	30.3	±	1.3	29.5	;	31.2	0.07	-							
**T1**	12	35.2	±	2.2	33.8	;	36.6	11	35.6	±	2.2	34.1	;	37.1	0.18	s							
***change***	*12*	***5.0***	***±***	***1.8***	***3.9***	***;***	***6.2***	*11*	***5.3***	***±***	***1.6***	***4.2***	***;***	***6.3***	*0.18*	*s*							
**Archperimeter**																	0.879	0.745	0.537	**< 0.001**	0.547	0.455	0.709
**T0**	12	66.2	±	2.5	64.6	;	67.7	11	66.2	±	2.4	64.6	;	67.9	0.00	-							
**T1**	12	70.2	±	4.0	67.7	;	72.8	11	69.5	±	4.4	66.6	;	72.5	0.17	s							
***change***	*12*	***4.1***	***±***	***2.7***	***2.4***	***;***	***5.7***	*11*	***3.3***	***±***	***2.4***	***1.6***	***;***	***4.9***	*0.31*	*s*							
**Crowding**																	0.762	0.532	0.583	**< 0.001**	0.493	0.229	0.197
**T0**	12	−7.1	±	2.3	−8.5	;	−5.6	11	−6.6	±	2.3	−8.1	;	−5.0	0.22	s							
**T1**	12	−2.8	±	1.9	−4.0	;	−1.6	11	−2.9	±	2.3	−4.5	;	−1.4	0.05	-							
***change***	*12*	***4.3***	***±***	***2.0***	***3.0***	***;***	***5.5***	*11*	***3.4***	***±***	***1.7***	***2.3***	***;***	***4.5***	*0.48*	*s*							
**COWR**																	0.398	0.089	0.271	**< 0.001**	0.107	0.562	0.608
**T0**	12	46.0	±	6.9	41.6	;	50.4	11	45.4	±	7.0	40.7	;	50.0	0.09	-							
**T1**	12	54.9	±	8.2	49.7	;	60.1	11	59.5	±	5.4	55.9	;	63.1	0.66	sm							
***change***	*12*	***8.9***	***±***	***6.7***	***4.6***	***;***	***13.2***	*11*	***14.1***	***±***	***6.6***	***9.6***	***;***	***18.5***	*0.78*	*sm*							
**COWL**																	0.079	0.059	0.070	**< 0.001**	**0.024**	0.775	0.514
**T0**	12	44.0	±	4.7	41.0	;	47.0	11	44.4	±	6.8	39.8	;	48.9	0.07	-							
**T1**	12	52.2	±	7.0	47.7	;	56.6	11	59.1	±	6.8	54.5	;	63.7	1.00	m							
***change***	*12*	***8.2***	***±***	***4.9***	***5.0***	***;***	***11.3***	*11*	***14.7***	***±***	***6.8***	***10.2***	***;***	***19.3***	*1.11*	*ml*							
**Age**																					**< 0.001**	0.291	0.378
**T0**	12	8.1	±	0.5	7.8	;	8.4	11	7.8	±	0.7	7.3	;	8.2	0.50	s							
**T1**	12	9.8	±	0.5	9.5	;	10.2	11	10.1	±	0.6	9.7	;	10.5	0.55	s							
***change***	*12*	***1.8***	***±***	***0.4***	***1.5***	***;***	***2.0***	*11*	***2.3***	***±***	***0.2***	***2.2***	***;***	***2.5***	*1.56*	*l*							

IC: intercanine distance; IPM: interpremolar distance; IM: intermolar distance; COW: curve of Wilson, right R, left L. Effect size according to Wahlsten (2011): -, no difference; s, small; sm, small-medium; m, medium; ml, medium-large; l, large. Prot: main effect of protocol (1 vs. 2); Sex: main effect of sex (M vs. F); Time: main effect of time (T0 vs. T1); P*S: interaction Protocol-by-Sex; P*T: interaction Protocol-by-Time; S*T: interaction Sex-by-Time; P*S*T: interaction Protocol-by-Sex-by-Time. Significant effects are highlighted in bold.

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
