# Peer review of "Comparison of Mandibular Arch Expansion by the Schwartz Appliance Using Two Activation Protocols: A Preliminary Retrospective Clinical Study"

_jfmk, 2020, doi:10.3390/jfmk5030061_

Round 1

Reviewer 1 Report

For the authors:
1 - line 27: "The measurements of parameters such as intercanine distance IC)," - a parenthesis is missing before the acronym IC.

2 - Were the individuals chosen at random to integrate each group? it is not referred to in the paper.

3 - Was a sample size calculation tool used to arrive at the total number of individuals needed to observe differences between groups?

4 - Table 1: There is an excess number "1" in the table header.

5 - Line 288: "our protocol is more effective as it allows clinicians to obtain the desired results more quickly (...)". Although the authors assume that protocol 1 seems to have better results, it is not discussed why changing devices every 4 months during all the treatment period is capable of improving results.

6 - In the conclusion, the authors do not state whether they accept or reject the null hypothesis (described in the introduction, line 73).

Author Response

Point 1:

Point taken – see line 27

Point 2:

Point taken – see line 87-89

Point 3:

Yes. The sample size of 11 vs 12 subjects allowed us to assess non inferiority of Protocol 1 vs Protocol 2, with pairwise alpha = 0.0125 (corresponding to an experimentwise alpha = 0.05 with Bonferroni's correction for 4 comparisons), power = 0.80 and non-inferiority limit = 1.3 SD (corresponding to a medium-large effect size according to Wahlsten, 2011).  Sample size calculation was performed using Sealed Envelope Ltd. 2012. Power calculator for continuous outcome non-inferiority trial. [Online] Available from: https://www.sealedenvelope.com/power/continuous-noninferior/ [Accessed on Monday 27 Jan 2020]

Point 4:

Point taken – see line 206

Point 5:

Point taken – see line 156-158

Point 6:

Point taken – see line 303-304

Reviewer 2 Report

Dear Editor, 

I very much appreciate the opportunity you have given to me to review the paper entitled: " Comparison of mandibular arch expansion by the Schwartz appliance using two activation protocols: a preliminary retrospective clinical study.”

This is a paper which looked at an interesting aspect in orthodontics: the orthodontic treatment for mandibular arch expansion.

The paper is well designed and written, and the topic is important and relevant. However, few minor issues need to be clarified before considering for publication in “Journal of Functional Morphology and Kinesiology”

Quote: “The protocol followed by group 1 involved the doctor activating the appliance of 2/4 turn every 15 days and the replacement of the device every 4 months during all the treatment period even if the screw had not reached its expansion limit. The group 2 protocol required the activation of the appliance of a 1/4 turn per week by the parents of the patient and the use of the same device for the whole period”

Add a paragraph in the discussion section on the cost-effectiveness of the two clinical protocols and on the different compliance required by the patients and parents.

Quote: “The parameters were measured on dental casts at the beginning of the treatment (T0) and at the end (T1) by the same operator with a manual caliber to eliminate interexaminer errors. Measurements were repeated one month later by the same operator in order to avoid intraexaminer errors”.   Report the inter-and intra-examiner error of method (ICC) coefficients.

Explain in detail the methods and tools used for the measurements on dental casts and in particular of the COW. So as to improve the repeatability of your research protocol.

Report the significant value in the table (p-value and *).

Improve the captions of the graphs at the end of the paper and cite them in the text.

Merge figure 2 and 3 and report the measurement descriptions in the figure caption.

Were the patients affected by maxillary transverse deficiency? Have concurrent treatments been carried out on the upper arch?

Remove references number 30 and 34. They are not appropriate.

Author Response

Point 1

Add a paragraph in the discussion section on the cost-effectiveness of the two clinical protocols and on the different compliance required by the patients and parents.

Point taken – see line 224-227

Point 2

Report the inter and intra-examiner error of method (ICC) coefficients

We are not able to calculate the inter-examiner error of method coefficients since the measurements were made by the same operator in order to avoid this kind of mistake. For the intra-examiner error of method coefficient see line 104-108

Point 3

Explain in detail the method and tools used for the measurements on dental casts and in particular of the COW.

Point taken – see line 145-147

Point 4

Report the significant value in the table (p-value and *)

Point taken. We modified the table by inserting all the significance of ANOVA, modified the header of the table and inserted a further legend. We were unable to insert the asterisks next to the significant valued due to the lack of space, so we highlighted them in bold as explained in the legend.

Point 4

Improve the captions of the graphs at the end of the paper and cite them in the text.

Point taken – see the captions and line 196-198

Point 5

Merge figure 2 and 3 and report the measurement descriptions in the figure caption.

Point taken

Point 6

Were the patiens affected by maxillary transverse deficiency? Have concurrent treatments been carried out on the upper arch?

Point taken – see line 152

Point 7

Remove reference number 30 and 34.

Point taken